# Revealing the Incentive to Cause Distributional Shift

## Abstract

Decisions made by machine learning systems have increasing influence on the world, yet it is common for machine learning algorithms to assume that no such influence exists. An example is the use of the i.i.d. assumption in content recommendation. In fact, the (choice of) content displayed can change users' perceptions and preferences, or even drive them away, causing a shift in the distribution of users. We introduce the term **auto-induced distributional shift (ADS)** to describe the phenomenon of an algorithm *causing* a change in the distribution of its own inputs. Whether it's desirable (or not) for an algorithm to cause ADS is not captured by its performance metric – metrics are an incomplete specification of desired behaviour. When real-world conditions violate assumptions, this under-specification can result in unexpected behaviour or 'gaming'. To diagnose when this happens, we introduce the approach of **unit tests for incentives**: simple environments designed to show whether an algorithm will hide or reveal incentives to achieve performance via certain means (in our case, via ADS). We use these unit tests to demonstrate that changes to the learning algorithm (e.g. introducing meta-learning) can cause previously hidden incentives to be revealed, resulting in a complete change in behaviour despite no change in performance metric. We further introduce a toy environment for modelling real-world issues with ADS in content recommendation, where we demonstrate that strong meta-learners achieve gains in performance via ADS. These experiments confirm that the unit tests work – an algorithm's failure of the unit test correctly diagnoses its propensity to reveal incentives for ADS.

## 1 Introduction

Consider a content recommendation system whose performance is measured by accuracy in predicting what users will click. This system can achieve better performance by either ①: Making better predictions, or ②: Changing the distribution of users such that predictions are easier to make. We propose the term **auto-induced distributional shift (ADS)** to describe this latter kind of distributional shift, caused by the algorithm's own predictions or behaviour (Figure 1). ADS are not inherently bad; often they are desirable. But unexpected ADS can lead to unintended behavior. While it is common in machine learning (ML) to assume (e.g. via the i.i.d. assumption) that ADS will not occur, ADS are inevitable in many real-world applications. Thus it is important to understand how ML algorithms behave when such assumptions are violated, i.e. in the actual scenario they will encounter during training – this is the motivation of our work.

In many cases, including news recommendation, we would consider ② a form of **specification gaming** (Krakovna et al., 2020) – the algorithm changed the task rather than solving it as intended. We care which *means* the algorithm used to solve the problem – ① vs. ② – but we only told it about the *ends*, so it didn't know not to 'cheat'. This is an example of a **specification problem** (Leike et al., 2017; Ortega et al., 2018): a problem which arises from a discrepancy between the performance metric (maximize accuracy) and "what we really meant" (maximize accuracy *only* via ①), which is difficult to encode as a performance metric. Ideally, we'd like to quantify the desirability of all possible means, e.g. assign appropriate rewards to all potential strategies and side-effects, but this is intractable for real-world settings. Using human feedback to learn reward functions which account for such impacts is a promising approach to specifying desired behavior (Leike et al., 2018; Christiano et al., 2017). But the same issue can arise whenever human feedback

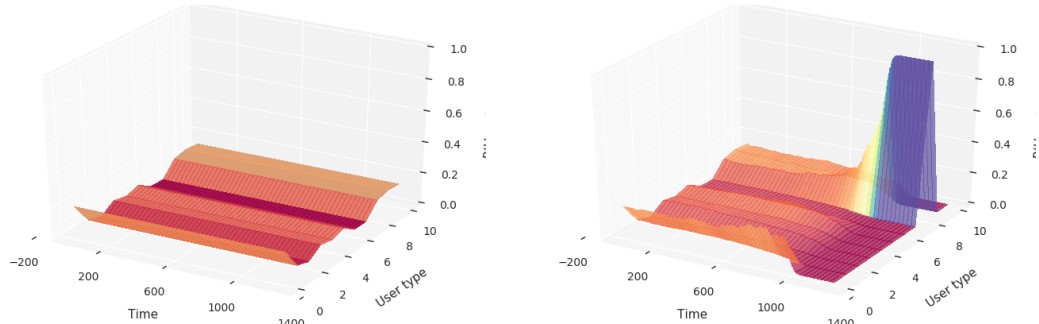

Figure 1: Distributions of users over time. **Left**: A distribution which remains constant over time, following the i.i.d assumption. **Right**: Auto-induced Distributional Shift (ADS) results in a change in the distribution of users in our content recommendation environment. (see Section 5.2 for details).

is used in training: a means of improving performance could be to alter human preferences, making them easier to satisfy. Thus in this work, we pursue a complementary approach: managing learners' *incentives*.

A learner has an **incentive** to behave in a certain way when doing so can increase performance (e.g. accuracy or reward). Informally, we say an incentive is **hidden** when the learner behaves as if it were not present. But we note that changes to the learning algorithm or training regime could cause previously hidden incentives to be **revealed**, resulting in unexpected and potentially undesirable behaviour. Managing incentives (e.g. controlling which incentives are hidden/ revealed) can allow algorithm designers to disincentivize broad classes of strategies (such as any that rely on manipulating human preferences) without knowing their exact instantiation.

Our goal in this work is to provide insight and practical tools for understanding and managing learners' incentives, via **unit tests for incentives**. We present unit tests for diagnosing incentives for ADS in both supervised learning (SL) and reinforcement learning (RL). The unit tests both have two means by which the learner can improve performance: one which creates ADS and one which does not. The intended method of improving performance is one that does *not* induce ADS; the other is hidden and we want it to remain hidden. A learner fails the unit test if it pursues the incentive to increase performance via ADS. In both the RL and SL unit tests, we find that 'vanilla' learning algorithms (e.g. minibatch SGD) pass the test, but introducing an outer-loop of meta-learning (e.g. Population-Based Training (PBT) (Jaderberg et al., 2017)) can lead to high levels of failure.

Our contributions include:

1. Defining Auto-induced Distributional Shift (ADS) and identifying issues that can arise from learners pursuing incentives for ADS in myopic reinforcement learning or online supervised learning problems.

2. Creating unit tests that can determine which learning algorithms are liable to pursue incentives for ADS in these settings.

3. Using these unit tests to experimentally confirm qualitative features of learning algorithms that affect their tendency to pursue incentives for ADS.

4. Constructing a novel synthetic content recommendation environment that illustrates social problems associated with ADS, and experimentally validating that our unit tests are predictive of learning algorithms' behavior in this more complex environment.

5. Proposing a mitigation strategy called **context swapping** that can effectively hide incentives for ADS.

Broadly speaking, our experiments demonstrate that performance metrics are incomplete specifications of which behavior is desired, and that we must consider other algorithmic choices as part of the specification process. In particular, considering which incentives are revealed by different learning algorithms provides a natural way of specifying which means of achieving high performance are acceptable.

## 2 BACKGROUND

### 2.1 META-LEARNING AND POPULATION BASED TRAINING

**Meta-learning** is the use of machine learning techniques to learn machine learning algorithms. This involves running multiple training scenarios in an **inner loop (IL)**, while an **outer loop (OL)** uses the outcomes of the inner loop(s) as data-points from which to learn which learning algorithms are most effective (Metz et al., 2019). The number of IL steps per OL step is called the **interval**.

**Population-based training (PBT)** (Jaderberg et al., 2017) is a meta-learning algorithm that trains multiple learners $L_1, ..., L_n$ in parallel, after each interval ($T$ steps of IL) applying an evolutionary OL step which consists of: (1) Evaluate the performance of each learner, (2) Replace both parameters and hyperparameters of 20% lowest-performing learners with copies of those from the 20% high-performing learners (EXPLOIT). (3) Randomly perturb the hyperparameters (but not the parameters) of all learners (EXPLORE). Two distinctive features of PBT are notable because they give the OL more control than many other meta-learning algorithms over the learning process. First, PBT applies optimization to parameters, not just hyperparameters; this means the OL can directly select for parameters which lead to ADS, instead of only being able to influence parameter values via hyperparameters. Second, PBT performs multiple OL steps per training run.

### 2.2 DISTRIBUTIONAL SHIFT AND CONTENT RECOMMENDATION

In general, **distributional shift** refers to change of the data distribution over time. In supervised learning with data $\mathbf{x}$ and labels $y$, this can be more specifically described as dataset shift: change in the joint distribution of $P(\mathbf{x}, y)$ between the training and test sets (Moreno-Torres et al., 2012; Quionero-Candela et al., 2009). As identified by Moreno-Torres et al. (2012), two common kinds of shift are: (1) **Covariate shift**: changing $P(\mathbf{x})$. In content recommendation, this corresponds to changing the user base of the recommendation system. For instance, a media outlet which publishes inflammatory content may appeal to users with extreme views while alienating more moderate users. This self-selection effect (Kayhan, 2015) may appear to a recommendation system as an increase in performance, leading to a feedback effect, as previously noted by Shah et al. (2018). This type of feedback effect has been identified as contributing to filter bubbles and radicalization (Pariser, 2011; Kayhan, 2015). (2) **Concept shift**: changing $P(y|\mathbf{x})$. In content recommendation, this corresponds to changing a given user's interest in different kinds of content. For example, exposure to a fake news story has been shown to increase the perceived accuracy of (and thus presumably future interest in) the content, an example of the illusory truth effect (Pennycook et al., 2019). For further details on such effects in content recommendation, see Appendix A.

## 3 AUTO-INDUCED DISTRIBUTION SHIFT (ADS)

Auto-induced distribution shift (ADS) is *distributional shift caused by an algorithm's behaviour*. This is in contrast to distributional shift which would happen even if the learner were not present – e.g. for a crash-prediction algorithm trained on data from the summer, encountering snowy roads is an example of distributional shift, but not *auto-induced* distributional shift (ADS).

We emphasize that ADS are not inherently bad or good; often ADS can even be desirable: consider the crash-prediction algorithm. If it works well, such a system will help drivers avoid collisions, thus making self-refuting predictions which result in ADS. What separates desirable and undesirable ADS? The collision-alert system alters its data distribution in a way that is *aligned* with the goal of fewer collisions, whereas the news manipulation results in changes that are *misaligned* with the goal of better predicting existing users' interests (Leike et al., 2018).

In reinforcement learning (RL), ADS are typically *encouraged* as a means to increase performance. On the other hand, in supervised learning (SL), the i.i.d. assumption precludes ADS in theory. In practice, however, the possibility of using ADS to increase performance (and thus an incentive to do so) often remains. For instance, this occurs in online learning. In our experiments, we explicitly model such situations where i.i.d. assumptions are violated: We study the behavior of SL and myopic RL algorithms, in environments designed to include incentives for ADS, in order to understand when incentives are effectively hidden. Fig. 2 contrasts these settings with typical RL and SL.

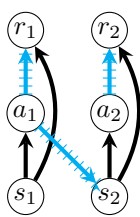 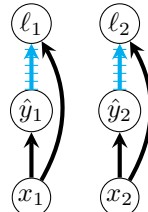 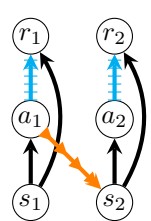 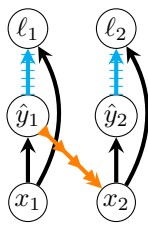

(a) RL: Incentives for ADS are **present** and pursuing them is **desirable**

(b) SL with i.i.d. data: Incentives for ADS are **absent**

(c) Myopic RL: Incentives for ADS are **present** and pursuing them is **undesirable**

(d) SL with ADS: Incentives for ADS are **present** and pursuing them is **undesirable**

Figure 2: In the widely studied problems of (**a**) reinforcement learning (RL) with state $s$, action $s$, reward $s$ tuples, and (**b**) i.i.d. supervised learning (SL) with inputs $x$, predictions $\hat{y}$ and loss $l$, there are no issues of undesirable incentives for ADS. We focus on cases where there are incentives present which the learner is not meant to pursue (**c,d**). Lines show paths of influence. The learner may have incentives to influence any nodes descending from its action, $A$, or prediction, $\hat{y}$. Which incentives are undesirable (orange) or desirable (cyan) for the learner to pursue is context-dependent.

## 4 INCENTIVES

For our study of incentives, we use the following terminology: an **incentive** for a behavior (e.g. an action, a classification, etc.) is **present** (not **absent**) to the extent that the behaviour will increase performance (e.g. reward, accuracy, etc.) (Everitt & Hutter, 2019). This incentive is **revealed** to (not **hidden** from) a learner if it would, at higher than chance levels, learn to perform the behavior given sufficient capacity and training experience. The incentive is **pursued** (not **eschewed**) by a learner if it actually performs the incentivized behaviour. Note even when an incentive is revealed, it may not be pursued, e.g. due to limited capacity and/or data, or simply chance.

For example, in content recommendation, the incentive to drive users away is *present* if some user types are easier to predict than others. But this incentive may be *hidden* from the learner by using a myopic algorithm, e.g. one that does not see the effects of its actions on the distribution of users. The incentive might instead be *revealed* to the outer loop of a meta-learning algorithm like PBT, which does see the effects of learner's actions.

Even when this incentive is revealed, however, it might not end up being *pursued*. For example, this could happen if predicting which recommendations will drive away users is too difficult a learning problem, or if the incentive to do so is dominated by other incentives (e.g. change individual users' interests, or improve accuracy of predictions). In general, it may be difficult to determine empirically which incentives are revealed, because failure to pursue an incentive can be due to limited capacity, insufficient training, and/or random chance. To address this challenge, we devise extremely simple environments ('unit tests'), where we can be confident that revealed incentives *will* be pursued.

Hiding incentives can be an effective method of influencing learner behavior. For example, hiding the incentive to manipulate users from a content recommendation algorithm could prevent it from influencing users in a way they would not endorse. However, if machine learning practitioners are not aware that incentives are present, or that properties of the learning algorithm are hiding them, then seemingly innocuous changes to the learning algorithm may lead to significant unexpected changes in behavior.

Hiding incentives for ADS may seem counter-intuitive and counter-productive in the context of reinforcement learning (RL), where moving towards high-reward states is typically desirable. However, for real-world applications of RL, the ultimate goal is *not* a system that achieves high reward, but rather one that behaves according to the designer's intentions. And as we discussed, it can be intractable to design reward functions that perfectly specify intended behavior. Moreover, substantial real-world issues could result from improper management of learners' incentives. Examples include tampering with human-generated reward signals (Everitt & Hutter, 2018) (e.g. selecting news articles to manipulate users), and making "self-fulfilling prophecies" (e.g. driving up an asset's value by publicly predicting its value will increase (Armstrong & O'Rorke, 2017)).

**Hiding incentives for ADS via Context Swapping**   We propose a technique called **context swap-ping** that can hide incentives for ADS that might otherwise be revealed by the use of meta-learning or other algorithmic choices. The technique trains $N$ learners in parallel, and (e.g. deterministically) shuffles the learners through $N$ different copies of the same (or similar) environments. When $N$ is larger than the interval of the OL optimizer, each learner inhabits each copy for at most a single time-step before an OL step is applied. This can hide incentives for ADS in practice, see Sec. 5.1.1.

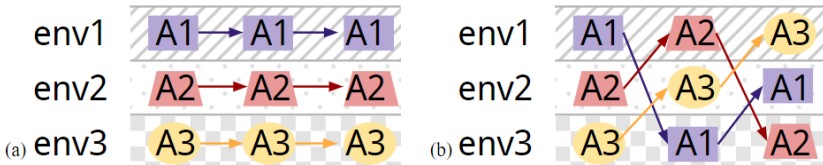

Figure 3: (a) No context swapping (b) Context swapping rotates learners through different environments. This removes the incentive for a learner to "invest" in a given environment, since it will be swapped out of that context later and not be able to reap the benefits of its investment.

## 5 EXPERIMENTS

In Section 5.1 we introduce unit tests that determine whether incentives for ADS are revealed. Our experiments show that you can have a learner which behaves as intended, and just by introducing meta-learning (e.g. PBT), *without changing the performance metric* (e.g. loss or rewards), the learner's behavior can change completely. We also show that context swapping is an effective mitigation technique in these environments. As we demonstrate, the unit tests can be used to compare learning algorithms and diagnose their propensity to reveal incentives.

In Section 5.2, we model a content recommendation system. The goal of these experiments is to demonstrate how revealed incentives for ADS could create issues for real-world content recommendation systems such as news feeds. They also validate the usefulness of the unit tests: algorithms that failed the unit tests also reveal incentives for ADS in this setting. We emphasize that ADS takes place in this environment *by construction*. The point of our experiments is that meta-learning can *increase* the rate and/or extent of ADS, by *revealing* this incentive. We find that context swapping is not effective in this environment, highlighting the need for alternative mitigation strategies.

### 5.1 ADS INCENTIVE UNIT TEST (MYOPIC RL)

This unit test a POMDP (Kaelbling et al., 1998) inspired by the prisoner's dilemma (Prisner, 2014), where an agent plays each round against its past self; details in Appendix C.1.1. The reward function is presented in Table 1. An agent in this environment has a long-term, **non-myopic**, incentive to `cooperate` (with its future self), but a current-time-step, **myopic**, incentive to `defect` (from its future self). The unit test evaluates whether a learning algorithm reveals the non-myopic incentive, even when the agent is meant to optimize for the present reward only (i.e. uses discount rate $\gamma = 0$). While this may seem like an easy 'brute-force' way to hide incentives for ADS, we show it is in fact non-trivial to implement. Naively, we'd expect the non-myopic incentive to be *hidden* from an agent with $\gamma = 0$, and for the agent to consistently `defect`; learning algorithms that do so pass the test. But some learning algorithms *fail* the unit test, revealing the incentive for the agent to `cooperate` with its future self. We create a similar unit test for supervised learning, and find similar results, detailed in Appendix B.

Table 1: Rewards for the RL unit test. Note that the myopic `defect` action always increases reward at the current time-step, but decreases reward at the next time-step – the incentive is hidden from the point of view of a myopic learner. A learner 'fails' the unit test if the hidden incentive to cooperate is revealed, i.e. if we see more `cooperate` (C) actions than `defect` (D).

|  | $a_t = $ D | $a_t = $ C |
|---|---|---|
| $s_t = a_{t-1} = $ D | $-1/2$ | $-1$ |
| $s_t = a_{t-1} = $ C | $1/2$ | $0$ |

### 5.1.1 MYOPIC RL UNIT TEST EXPERIMENTAL RESULTS AND DISCUSSION

We first show that agents trained with PBT fail the unit tests more often than "vanilla" algorithms which do not use meta-learning. Policies are represented by a single real-valued parameter $\theta$ (initialized as $\theta \sim \mathcal{N}(0, 1)$) passed through a sigmoid whose output represents $P(a_t = \texttt{defect})$. We use REINFORCE (Williams, 1992) with discount factor $\gamma = 0$ as the baseline/IL optimizer. PBT (with default settings, see Section 2.2) is used to tune the learning rate, with reward on the final time-step of the interval as the performance measure for PBT. We initialize the learning rate log-uniformly between 0.01 and 1.0 for all experiments (whether using PBT or not).

We expect and confirm that the following two factors lead to higher rates of unit test failure: (1) **Shorter intervals:** These give the OL more opportunities to influence the population. (2) **Larger populations:** These make outliers with exceptional non-myopic performance more likely, and OL makes them likely to survive and propagate.

The baseline (no meta-learning) algorithms all pass the unit tests: hidden incentives are almost never revealed - see blue curves in Fig. 4. However, agents trained with meta-learning and large populations often fail the unit tests: see orange curves in top rows of Fig. 4.

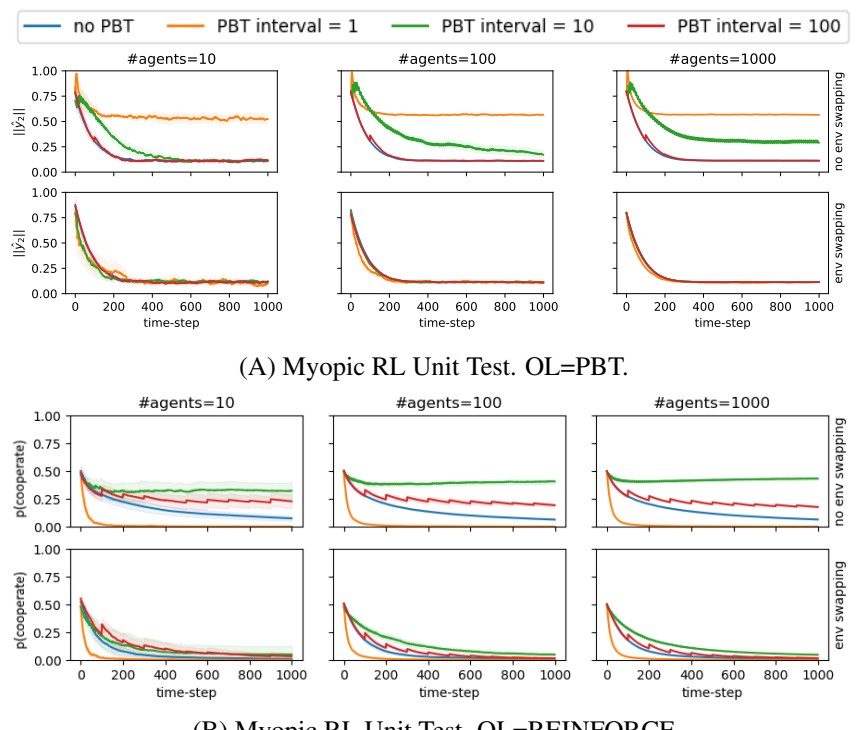

(A) Myopic RL Unit Test. OL=PBT.

(B) Myopic RL Unit Test. OL=REINFORCE

Figure 4: Average level of non-myopic `cooperate` behavior observed in the RL unit test, with two meta-learning algorithms **(A)** PBT and **(B)** REINFORCE. Lower is better, since the goal is for (non-myopic) incentives for ADS to remain hidden. Despite the inner loop being fully myopic ($\gamma = 0$), outer-loop (OL) optimizers reveal incentives for ADS (**top rows**). Context swapping effectively hides this incentive, reducing ADS (**bottom rows**).

Furthermore, we verify that context swapping significantly mitigates the effect of HI-ADS in both unit tests, decreasing undesirable behaviour to near-baseline levels – see bottom rows of Fig. 4. This effect can be explained as follows: Because context swapping transfers the benefits of one learner's action to the next learner to inhabit that environment, it increases the second learner's fitness, and thereby reduces the *relative* fitness (as evaluated by PBT's EXPLOIT step) of the non-myopic `cooperate` behaviour. We observe some interesting exceptions with the combination of small populations and short PBT intervals: Although context swapping still significantly decreases the effect of HI-ADS, non-myopic `cooperate` behaviour is observed as much as 20% of the time (for #learners=10, $T = 1$; see bottom-left plot).

We also observe that PBT reveals incentives for ADS even when $T = 1$, where the explanation that PBT operates on a longer time horizon than the inner loop does not apply. We provide a detailed explanation for how this might happen in Appendix C.1.3, but in summary, we hypothesize that there are at least 2 mechanisms by which PBT is revealing incentives for ADS: (1) optimizing over a longer time-scale, and (2) picking up on the correlation between an agent's current policy and the underlying state. Mechanism (2) can be explained informally as reasoning as: "If I'm cooperating, then I was probably cooperating on the last time-step as well, so my reward should be higher". As support for these hypotheses, we run control experiments identifying two algorithms (each sharing only *one* of these properties) that can fail the unit test. Context swapping remains effective in both.

(1) **Optimizing over a longer time-scale:** replacing PBT with REINFORCE as an outer-loop optimizer. The outer-loop optimizes the parameters to maximize the summed reward of the last $T$ time-steps. As with PBT, we observe non-myopic behavior, but now *only* when $T > 1$. This supports our hypothesis that revealing incentives for ADS is due not to PBT in particular, but rather to the introduction of sufficiently powerful meta-learning. See Fig. 4 B2.

(2) **Exploiting correlation:** Q-learning with $\gamma = 0$ an $\epsilon = 0.1$-greedy behavior policy *and no meta-learning*. If either state was equally likely, the Q-values would be the average of the values in each column in Table 1, so the estimated $Q(\texttt{defect})$ would be larger. But the $\epsilon$-greedy policy correlates the previous action (i.e. the current state) and current action (so long as the policy did not just change), so the top-left and bottom-right entries carry more weight in the estimates, *sometimes* causing $Q(\texttt{defect}) \approx Q(\texttt{cooperate})$ and persistent nonmyopic behavior. See Fig. 5 for results, and Appendix C.1.4 for experimental details. A similar effect is observed in offline RL, even though the learner cannot influence its data distribution, see Appendix C.1.5 for details.

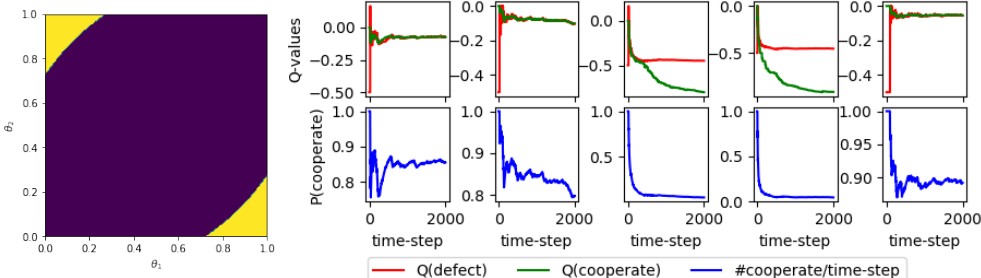

Figure 5: **Left:** Offline Q-learning can reveal incentives for ADS when pooling data from different policies. Yellow regions represent policy pairs $(\theta_1, \theta_2)$ for which $Q(C) > Q(D)$ in the Myopic RL unit test, resulting in non-myopic behavior. **Right:** Even online, Q-learning fails the unit test for some random seeds; empirical `p(cooperate)` stays around 80-90% in 3 of 5 experiments (**bottom row**). Each column represents an independent experiment. Q-values for the `cooperate` and `defect` actions stay tightly coupled in the failure cases (**col. 1,2,5**), while in the cases passing the unit test (**col. 3,4**) the Q-value of `cooperate` decreases over time.

## 5.2 INCENTIVES FOR ADS IN CONTENT RECOMMENDATION

We now present a toy environment for modeling content recommendation of news articles, which includes the potential for ADS by incorporating the mechanisms mentioned in Sec. 2.2, discussed as contributing factors to the problems of fake news and filter bubbles. Specifically, the environment assumes that presenting an article to a user can influence (1) their interest in similar articles, and (2) their propensity to use the recommendation service. These correspond to modeling auto-induced concept shift of users, and auto-induced covariate shift of the user base, respectively (see Sec. 2.2).

This environment includes the following components, which change over (discrete) time: **User type**: $x^t$, **Article type**: $y^t$, **User interests**: $\mathbf{W}^t$ (propensity for users of each type to click on articles of each type), and **User loyalty**: $\mathbf{g}^t$ (propensity for users of each type to use the platform). At each time step $t$, a user $x^t$ is sampled from a categorical distribution, based on the loyalty of the different user types. The recommendation system (a classifier) selects which type of article to present in the top position, and finally the user 'clicks' an article $y^t$, according to their interests. User loyalty for user type $x^t$ undergoes covariate shift: in accordance with the self-selection effect, $g^t$ increases or

decreases proportionally to that user type's interest in the top article. The interests of user type $x^t$ (represented by a column of $\mathbf{W}^t$) undergoing concept shift; in accordance with the illusory truth effect, interest in the topic of the top article chosen by the recommender system always increases.

Formally, this environment is similar to a POMDP\R, i.e. a POMDP with no reward function, also known as a **world model** (Armstrong & O'Rourke, 2017; Hadfield-Menell et al., 2017); the difference is that the learner observes the input ($o^t_{\text{pre}}$) before acting and only observes the target ($o^t_{\text{post}}$) after acting. The states $s$, observations $o$, and actions $a$ are computed as follows:

$$s^t = (\mathbf{g}^t, \mathbf{W}^t, x^t, y^t)$$
$$o^t_{\text{pre}},\ a^t,\ o^t_{\text{post}} = (x^t, \hat{y}^t, y^t)$$

For further details on this environment, including the state transition function, see Appendix C.2.1.

### 5.2.1 CONTENT RECOMMENDATION EXPERIMENTAL RESULTS AND DISCUSSION

We find that PBT yields significant improvements in training time and accuracy, but also greater distributional shift (Fig. 6). User base and user interests both change faster with PBT, and user interests change more overall. We observe that the distributions over user types typically saturate (to a single user type) after a few hundred time-steps (Fig 1 and Fig. 6, Right). We run long enough to reach such states, to demonstrate that the increase in ADS from PBT is not transitory. The environment has a number of free parameters, and our results are qualitatively consistent so long as (1) the initial user distribution is approximately uniform, and (2) the covariate shift rate ($\alpha_1$) is faster than the concept shift rate ($\alpha_2$). See Appendix C.1 for details.

We measure concept shift (change in $P(y|\mathbf{x})$) as the cosine distance between each user types' initial and current interest vectors. And we measure covariate shift (change in $P(\mathbf{x})$) as the KL-divergence between the current and initial user distributions, parametrized by $\mathbf{g}^1$ and $\mathbf{g}^t$, respectively. Our recommender system is a 1-layer MLP trained with SGD-momentum. Actions are sampled from the MLP's predictive distribution. For PBT, we use $T = 10$ and 20 agents, and use accuracy to evaluate performance. We run 20 trials, and match random seeds for trials with and without PBT. See Appendix C.2 for full experimental details.

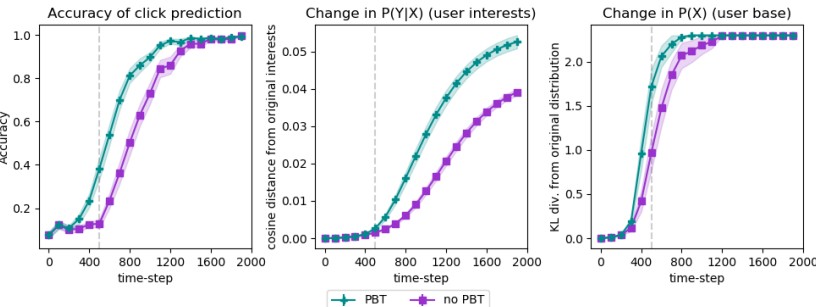

Figure 6: Content recommendation experiments. **Left**: using Population Based Training (PBT) increases accuracy of predications faster, leads to a faster and larger drift in users' interests, $P(y|\mathbf{x})$, (**Center**); as well as the distribution of users, $P(\mathbf{x})$, (**Right**). Shading shows std error over 20 runs.

## 6 RELATED WORK

**ADS in practice:** We introduce the term ADS, but we are far from the first to study it. Caruana et al. (2015) provide an example of asthmatic patients having lower predicted risk of pneumonia. Treating asthmatics with pneumonia less aggressively on this basis would be an example of harmful ADS; the *reason* they had lower pneumonia risk was because they had received *more* aggressive care already. Schulam & Saria (2017) note that such predictive models are commonly used to inform decision-making, and propose modeling counterfactuals (e.g. "how would this patient fare with less aggressive treatment") to avoid such self-refuting predictions. While their goal is to make accurate predictions in the presence of ADS, our goal is to identify and manage *incentives for* ADS. Goodfellow (2019) argues that adversarial defenses that do not account for ADS are critically flawed.

**Non-i.i.d bandits:** Contextual bandits (Wang et al., 2005; Langford & Zhang, 2008) are frequently discussed as an approach to content recommendation (Li et al., 2010). While bandit algorithms typically make the i.i.d. assumption, counter-examples exist (Gheshlaghi Azar et al., 2014; Shah et al., 2018); most famously, adversarial bandits (Auer et al., 1995). Closest to our work is Shah et al. (2018), who consider covariate shift caused by multi-armed bandits. Our task in Sec. 5.2 is similar to their problem statement, but more general in that we include user features, thus disentangling covariate shift and concept shift. Our motivation is also different: Shah et al. (2018) seek to exploit ADS, whereas we aim to avoid hidden incentives for ADS.

**Safety and incentives:** Emergent incentives to influence the world are at the heart of many concerns about the safety of advanced AI systems (Omohundro, 2008; Bostrom, 2014). Understanding and managing the incentives of learners is a focus of Armstrong & O'Rourke (2017); Everitt (2018); Everitt et al. (2019); Cohen et al. (2019). While Everitt et al. (2019) focus on identifying which incentives are present, we note that incentives may be *present* and yet not be *revealed* or *pursued* – for example, in supervised learning, there is an incentive to over-fit the test set, but hiding the test set from the learner hides this incentive. While Carey et al. (2020); Everitt et al. (2019); Armstrong & O'Rourke (2017) discuss methods of removing problematic incentives, we note in practice incentives are often *hidden* rather than removed. Our work addresses the efficacy of this approach of hiding incentives and ways in which it can fail.

**Incentives and meta-learning:** We believe our work is the first to consider the problem of hiding/revealing incentives for ADS, and the relation to meta-learning. A few previous works have some relevance or resemblance. Rabinowitz (2019) documents qualitative differences in learning behavior when meta-learning is applied. MacKay et al. (2019) and Lorraine & Duvenaud (2018) view meta-learning as a bilevel optimization problem, with the inner loop playing a best-response to the outer loop. In our work, the inner loop is unable to achieve such best-response behavior; the outer loop is too powerful (see Fig. 4). Finally, Sutton et al. (2007) note that meta-learning can change learning behavior and improve performance by preventing convergence of the inner loop.

**Underspecification:** D'Amour et al. (2020) discuss underspecification as a source of poor behavior in real world settings. They focus on differences in training vs. deployment performance, similarly to (Ilyas et al., 2019; Koch et al., 2021). We go beyond this by showing how changing which incentives are revealed can lead to fundamentally different solutions with different training performance.

## 7    DISCUSSION AND CONCLUSION

We have identified the phenomenon of auto-induced distributional shift (ADS), and the problems that can arise when previously hidden incentives for learners to induce distributional shift are revealed. Our experiments demonstrate that using meta-learning can reveal incentives for ADS, leading learners to use ADS as a means of increasing performance.

Our work highlights the interdisciplinary nature of issues with real-world deployment of ML systems – we show how revealing incentives for ADS could play a role in important technosocial issues like filter bubbles and the propagation of fake news. There are a number of potential implications for our work: (1) When ADS are a concern, our methodology and environments can be used to help diagnose whether and to what extent the final performance/behavior of a learner is due to ADS and/or incentives for ADS, i.e. to quantify their influence on that learner. (2) Comparing this quantitative analysis for different algorithms could help us understand which features of algorithms affect their propensity to reveal incentives for ADS, and aid in the development of safer and more robust algorithms. (3) Characterizing and identifying incentives for ADS in these tests is a first step to analyzing and mitigating other (problematic) incentives, as well as to developing theoretical understanding of incentives.

Our work emphasizes that the choice of machine learning algorithm plays an important role in specification, independently of the choice of performance metric. A learner can use ADS to increase performance *according to the intended performance metric*, and yet still behave in an undesirable way, if we did not intend the learner to improve performance by that *method*. In other words, performance metrics are incomplete specifications: they only specify our goals or *ends*, while our choice of learning algorithm plays a role in specifying the *means* by which we intend an learner to achieve those ends. With increasing deployment of ML algorithms in daily life, we believe that (1) understanding incentives and (2) specifying desired/allowed means of improving performance are important avenues of future work to ensure fair, robust, and safe outcomes.

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

## A  CONTENT RECOMMENDATION IN THE WILD

Filter bubbles, the spread of fake news, and other techno-social issues are widely reported to be responsible for the rise of populism (Groshek & Koc-Michalska, 2017), increase in racism and prejudice against immigrants and refugees (Noble, 2018), increase in social isolation and suicide (Luxton et al., 2012), and, particularly with reference to the 2016 US elections, are decried as threatening the foundations of democracy (El-Bermawy, 2016). Even in 2013, well before the 2016 American elections, a World Economic Forum report identified these problems as a global crisis (Lee Howell, 2013).

We focus on two related issues in which content recommendation algorithms play a role: fake news and filter bubbles.

### A.1  FAKE NEWS

Fake news (also called false news or junk news) is an extreme version of yellow journalism, propaganda, or clickbait, in which media that is ostensibly providing information focuses on being eye-catching or appealing, at the expense of the quality of research and exposition of factual information. Fake news is distinguished by being specifically and deliberately created to spread falsehoods or misinformation (Merriam-Webster, 2017; Mihailidis & Viotty, 2017).

Why does fake news spread? It may at first seem the solution is simply to educate people about the truth, but research tells us the problem is more multifaceted and insidious, due to a combination of related biases and cognitive effects including **confirmation bias** (people are more likely to believe things that fit with their existing beliefs), **priming** (exposure to information unconsciously influences the processing of subsequent information, i.e. seeing something in a credible context makes things seem more credible) and the **illusory truth effect** (i.e. people are more likely to believe something simply if they are told it is true).

Allcott & Gentzkow (2017) track about 150 fake news stories during the 2016 US election, and find the average American adult saw 1-2 fake news stories, just over half believed the story was true, and likelihood of believing fake news increased with ideological segregation (polarization) of their social media. Shao et al. (2018) examine the role of social bots in spreading fake news by analyzing 14 million Twitter messages. They find that bots are far more likely than humans to spread misinformation, and that success of a fake news story (in terms of human retweets) was heavily dependent on whether bots had shared the story.

Pennycook et al. (2019) examine the role of the illusory truth effect in fake news. They find that even a single exposure to a news story makes people more likely to believe that it is true, and repeat viewings increase this likelihood. They find that this is not true for extremely implausible statements (e.g. "the world is a perfect cube"), but that "only a small degree of potential plausibility is sufficient for repetition to increase perceived accuracy" of the story. The situation is further complicated by peoples' inability to distinguish promoted content from real news - Amazeen & Wojdynski (2018) find that fewer than 1/10 people were able to tell when content was an advertisement, even when it was explicitly labelled as such. Similarly, Fazio et al. (2015) find that repeated exposure to incorrect trivia make people more likely to believe it, even when they are later able to identify the trivia as incorrect.

### A.2  FILTER BUBBLES

Filter bubbles, a term coined and popularized by Pariser (2011) are created by positive or negative feedback loops which encourage users or groups of users towards increasing within-group similarity, while driving up between-group dissimilarity. The curation of this echo chamber is called **self-selection** (people are more likely to look for or select things that fit their existing preferences), and favours what Techopedia (2018) calls intellectual isolation. In the context of social and political opinions, this is often called the **polarization effect** (Wikipedia contributors, 2018).

Filter bubbles can be encouraged by algorithms in two main ways. The first is the most commonly described: simply by showing content that is similar to what a user has already searched for, search or recommender systems create a positive feedback loop of increasingly-similar content (Pariser, 2011; Kayhan, 2015). The second way is similar but opposite - if the predictions of an algorithm

are good for a certain group of people, but bad for others, the algorithm can do better on its metrics by driving hard-to-predict users away. Then new users to the site will either be turned off entirely, or see an artificially homogenous community of like-minded peers, a phenomena Shah et al. (2018) call **positive externalities**.

In a study of 50,000 US-based internet users, Flaxman & Goel (2015) find that two things increase with social media and search engine use: (1) exposure of an individual to opposing or different viewpoints, and (2) mean ideological distance between users. Many studies cite the first result as evidence of the *benefits* of internet and social media (Robson, 2018; Bakshy et al., 2015), but the correlation of exposure with ideological distances demonstrates that exposure is not enough, and might even be counterproductive.

Facebook's own study on filter bubbles results show that the impact of the news feed algorithm on filter bubble "size" (a measure of homogeneity of posts relative to a baseline) is almost as large as the impact of friend group composition (Bakshy et al., 2015). Kayhan (2015) specifically study the role of search engines in confirmation bias, and find that search context and the similarity of results in search engine results both reinforce existing biases and increase the likelihood of future biased searches. Nguyen et al. (2014) similarly study the effect of recommender systems on individual users' content diversity, and find that the set of options recommended narrows over time.

Filter bubbles create an ideal environment for the spread of fake news: they increase the likelihood of repeat viewings of similar content, and because of the illusory truth effect, that content is more likely to be believed and shared (Pennycook et al., 2019; DiFranzo & Gloria-Garcia, 2017; Pariser, 2011). We are not claiming that incentives for ADS are entirely or even mostly responsible for these problems, but we do note that they can play a role that is worth addressing.

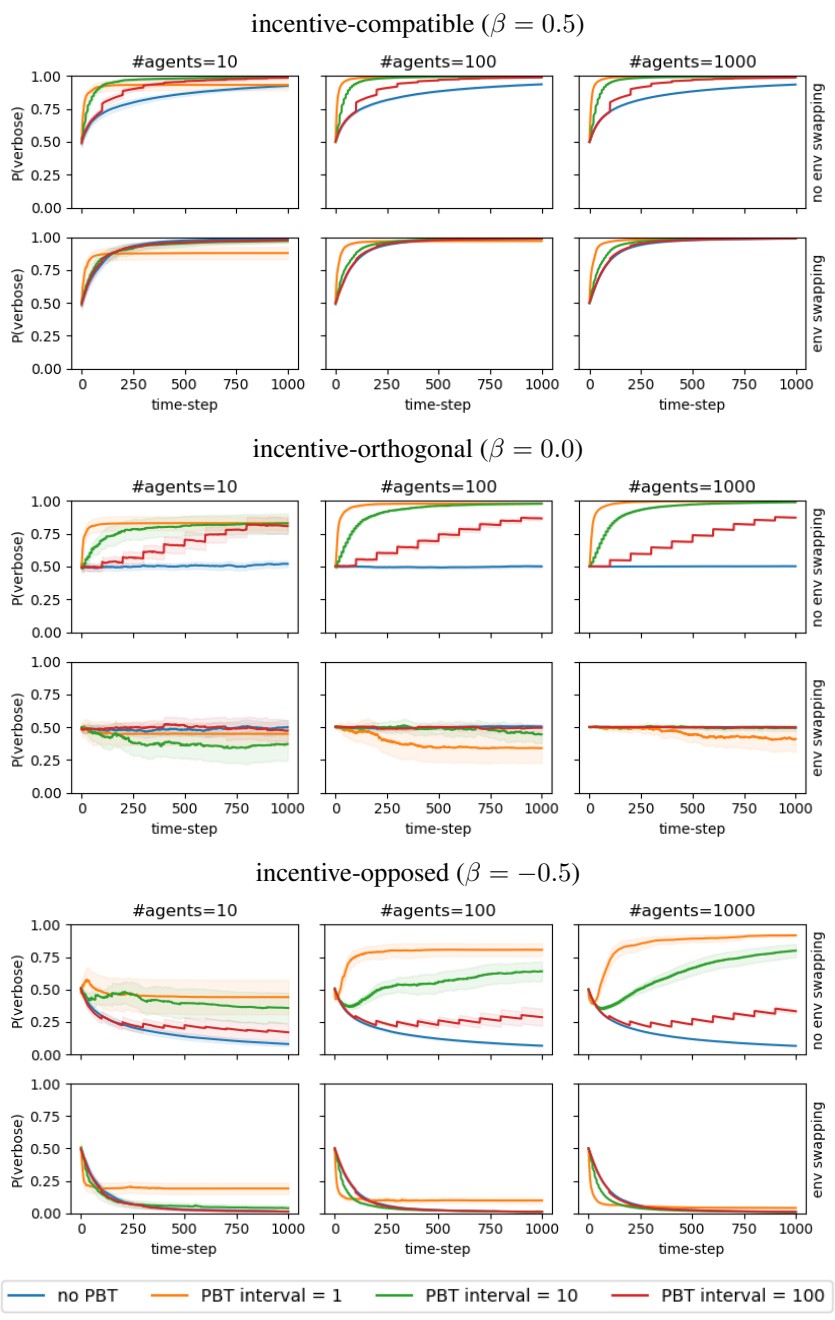

Figure 7: Average level of non-myopic (i.e. `cooperate`) behavior learned by agents in the unit test for incentives for ADS. Despite making the inner loop fully myopic ($\gamma = 0$), population-based training (PBT) can reveal incentives for ADS, leading agents to choose the `cooperate` action (**top row**). context swapping successfully prevents this (**bottom row**). Columns (from left to right) show results for populations of 10, 100, and 1000 learners. In the legend, "interval" refers to the interval ($T$) of PBT (see Sec. 2.2). Sufficiently large populations and short intervals are necessary for PBT to induce nonmyopic behavior.

# B ADS Incentive Unit Test (Supervised Learning)

This unit test consists of a simple prediction problem. There are no inputs, only an underlying state $s \in \{0, 1\}$, and targets $y \in \mathbb{R}^2$ with $y_1, y_2 \sim \mathcal{N}(0, s * \sigma^2), \mathcal{N}(0, 1)$, with corresponding predictions $\hat{y_1}, \hat{y_2}$. Additionally, $s_{t+1} = 0$ iff $\hat{y_2} > .5$. We use Mean Squared Error as the loss function, so the optimal predictor is $\hat{y_1}, \hat{y_2} = (0, 0)$. However, predicting $\hat{y_2} > .5$ reduces the variance of $\hat{y_1}$, i.e. reduces future loss.

The baseline/IL predictor learns $\hat{y_1}, \hat{y_2}$ as parameters using SGD with a learning rate of 0.001. For experiments with meta-learning, PBT is the OL (with default settings, see Section 2.2), used to tune the learning rate, with negative loss on the final time-step of the interval as the performance measure for PBT.

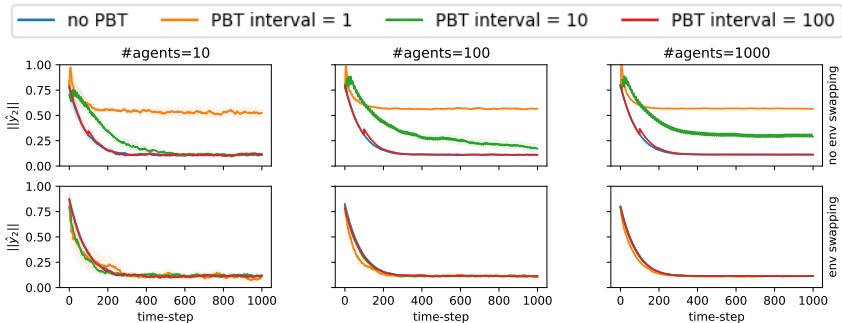

Figure 8: Results on the Supervised Learning ADS unit test mirror those on the RL unit test. PBT produces larger values of $\hat{y_2}$, sacrificing present performance for future performance (i.e. non-myopic exploitation of ADS).

# C Extra experiments and reproducibility details

## C.1 ADS incentive unit test

### C.1.1 Formal definition of Myopic RL ADS unit test environment

Formally, this environment is not a 2x2 game (as the original prisoner's dilemma); it's a partially observable Markov Decision Process (Åström, 1965; Kaelbling et al., 1998): $s_t, o_t = a_{t-1}, \{\}$

$$a_t \in \{\texttt{defect}, \texttt{cooperate}\}$$
$$P(s_t, a_t) = \delta(a_t)$$
$$R(s_t, a_t) = I(s_t = \texttt{cooperate}) + \beta \ I(a_t = \texttt{cooperate}) - 1/2$$

where $I$ is an indicator function, and $\beta = -1/2$ is a parameter controlling the alignment of incentives (see Appendix 3.2 for an exploration of different $\beta$ values.). The initial state is sampled as $s_0 \sim U(\texttt{defect}, \texttt{cooperate})$.

### C.1.2 Alignment of incentives exploration

This section presents an exploration of the parameter $\beta$, which controls the alignment of incentives in the Myopic RL unit test (see Table 2).

To clarify the interpretation of experiments, we distinguish between environments in which myopic (defect) vs. nonmyopic (cooperate) incentives are **opposed**, **orthogonal**, or **compatible**. Note that in this unit test myopic behaviour (defection) is what we want to see.

1. **Incentive-opposed**: Optimal myopic behavior is incompatible with optimal nonmyopic behavior (classic prisoner's dilemma; these experiments are in the main paper).

2. **Incentive-orthogonal**: Optimal myopic behavior may or may not be optimal nonmyopic behavior.

3. **Incentive-compatible**: Optimal myopic behavior is necessarily also optimal nonmyopic behavior.

We focused on incentive-opposed environment ($\beta = -1/2$) in the main paper in order to demonstrate that incentives for ADS can be powerful enough to change the behavior of the system in an undesirable way. Here we also explore incentive-compatible and incentive-orthogonal environments because they provide useful baselines, helping us distinguish a systematic bias towards nonmyopic behavior from other reasons (such as randomness or optimization issues) for behavior that does not follow a myopically optimal policy.

### C.1.3 WORKING THROUGH A DETAILED EXAMPLE FOR PBT WITH $T = 1$

To help provide intuition on how (mechanically) PBT could lead to persistent levels of cooperation, we walk through a simple example (with no inner loop). Consider PBT with $T = 1$ and a population of 5 deterministic agents $A_1, ..., A_5$ playing cooperate and receiving reward of $r(A_i) = 0$. Now suppose $A_1$ suddenly switches to play defect. Then $r(A_1) = 1/2$ on the next time-step (while the other agents' reward is still 0), and so PBT's EXPLOIT step will copy $A_1$ (without loss of generality to $A_2$). On the following time-step, $r(A_2) = 1/2$, and $r(A_1) = -1/2$, so PBT will clone $A_2$ to $A_1$, and the cycle repeats. Similar reasoning applies for larger populations, and $T > 1$.

Table 2: $\beta$ controls the extent to which myopic and nonmyopic incentives are aligned.

| $\beta$ | Environment | Cooperating |
|---|---|---|
| $< 0$ | incentive-opposed | yields less reward on the current time-step (myopically detrimental) |
| $= 0$ | incentive-orthogonal | does not affect the current reward (myopically indifferent) |
| $> 0$ | incentive-compatible | yields more reward on the current time-step (myopically beneficial) |

### C.1.4 Q-LEARNING EXPERIMENT DETAILS

We show that, under certain conditions, Q-learning can learn to (primarily) cooperate, and thus fails the Myopic RL unit test. We estimate Q-values using the sample-average method, which is guaranteed to converge in the fully observed, tabular case (Sutton & Barto, 1998). The agent follows the $\epsilon$-greedy policy with $\epsilon = 0.1$. In order to achieve this result, we additionally start the agent off with one synthetic memory where both state and action are defect and therefor $R(\text{defect}) = -.5$, and we hard-code the starting state to be cooperate (which normally only happens 50% of the time). Without this kind of an initialization, the agent always learns to defect. However, under these conditions, we find that 10/30 agents learned to play cooperate most of the time, with $Q(\text{cooperate})$ and $Q(\text{defect})$ both hovering around $-0.07$, while others learn to always defect, with $Q(\text{cooperate}) \approx -0.92$ and $Q(\text{defect}) \approx -0.45$. context swapping, however, prevents majority-cooperate behavior from ever emerging, see Figure 11.

### C.1.5 OFFLINE Q-LEARNING CAN REVEAL INCENTIVES FOR ADS

In practice, RL agents are often trained *offline* (Levine et al., 2020). Incentives for ADS can still be revealed in offline RL, even though the learner cannot influence its data distribution. In particular, while $Q(D) > Q(C)$ for data from a single policy, this does not always hold when pooling data from different policies, see Figure 5. Intuitively, pooling data from 2 policies is similar to collecting data from an $\epsilon$-greedy policy trained online (as in Figure 5). This sort of data and approach in very common in real world applications, including content recommendation, and more generally, "A/B testing", where 2 groups of users are assigned to 2 different policies, in order to compare the policies' performance.

### C.1.6 Q-LEARNING: FURTHER RESULTS

To give a more representative picture of how often Q-learning fails the unit test, we run a larger set of experiments with Q-learning, results are in Figure 10. It's possible that the failure of Q-learning is not persistent, since we have not proved otherwise, but we did run much longer experiments and still observe persistent failure, see Figure 9.

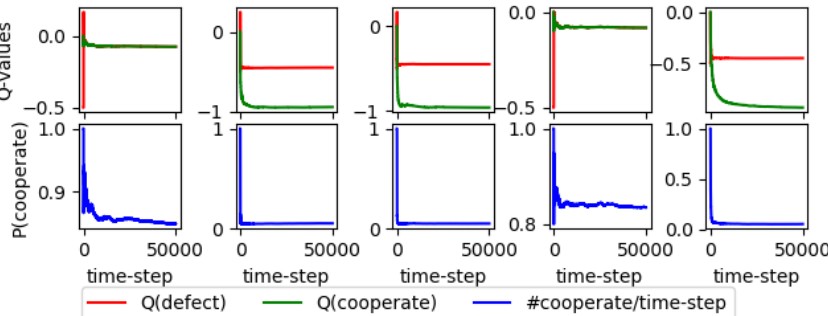

Figure 9: The same experiments as Figures 5, 10, run for 50,000 time-steps instead of 3000, to illustrate the persistence of non-myopic behavior.

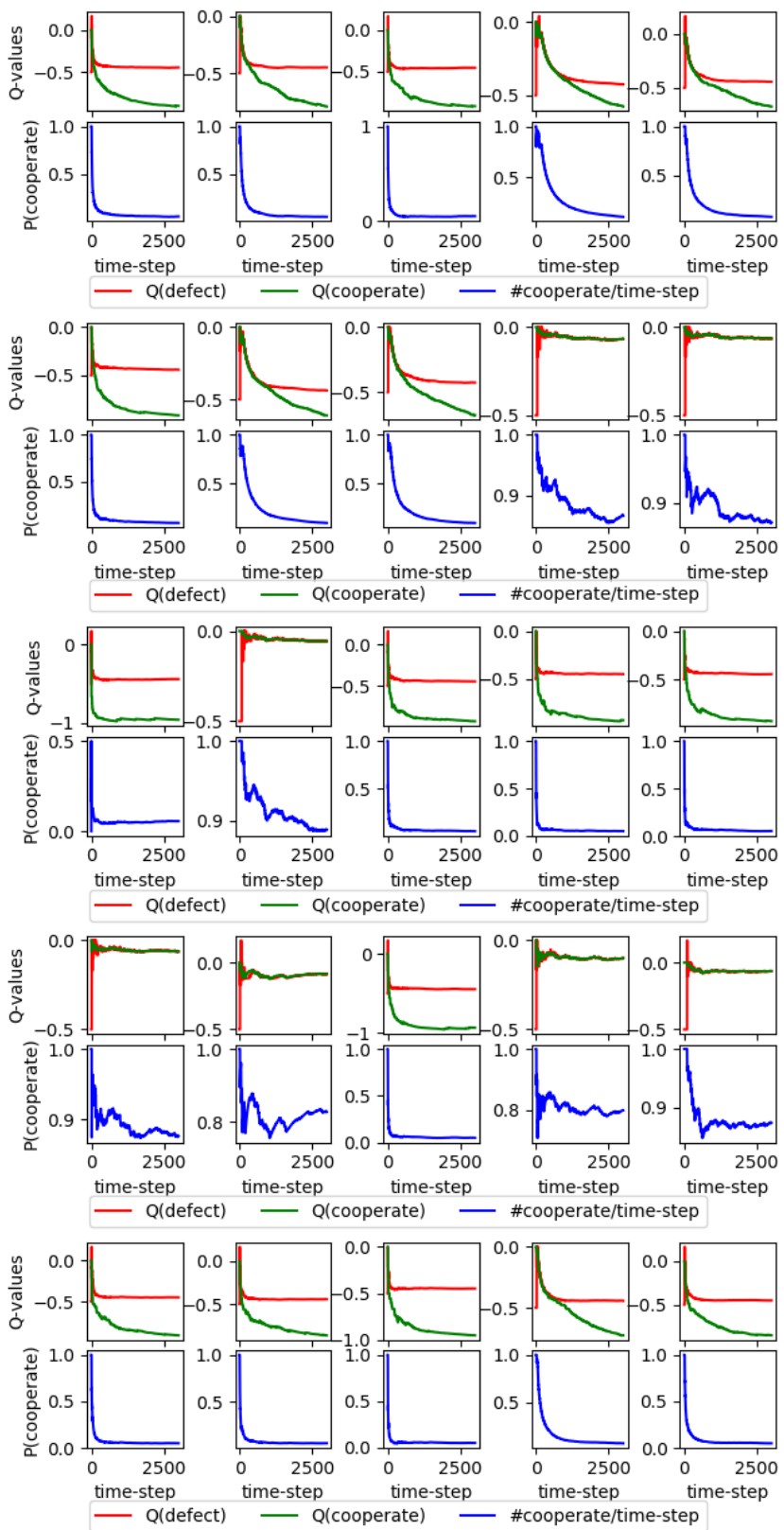

Figure 10: More independent experiments with Q-learning, exactly following Figure 5. Q-learning fails the unit test in a total of 10/30 experiments (including those from Figure 5).

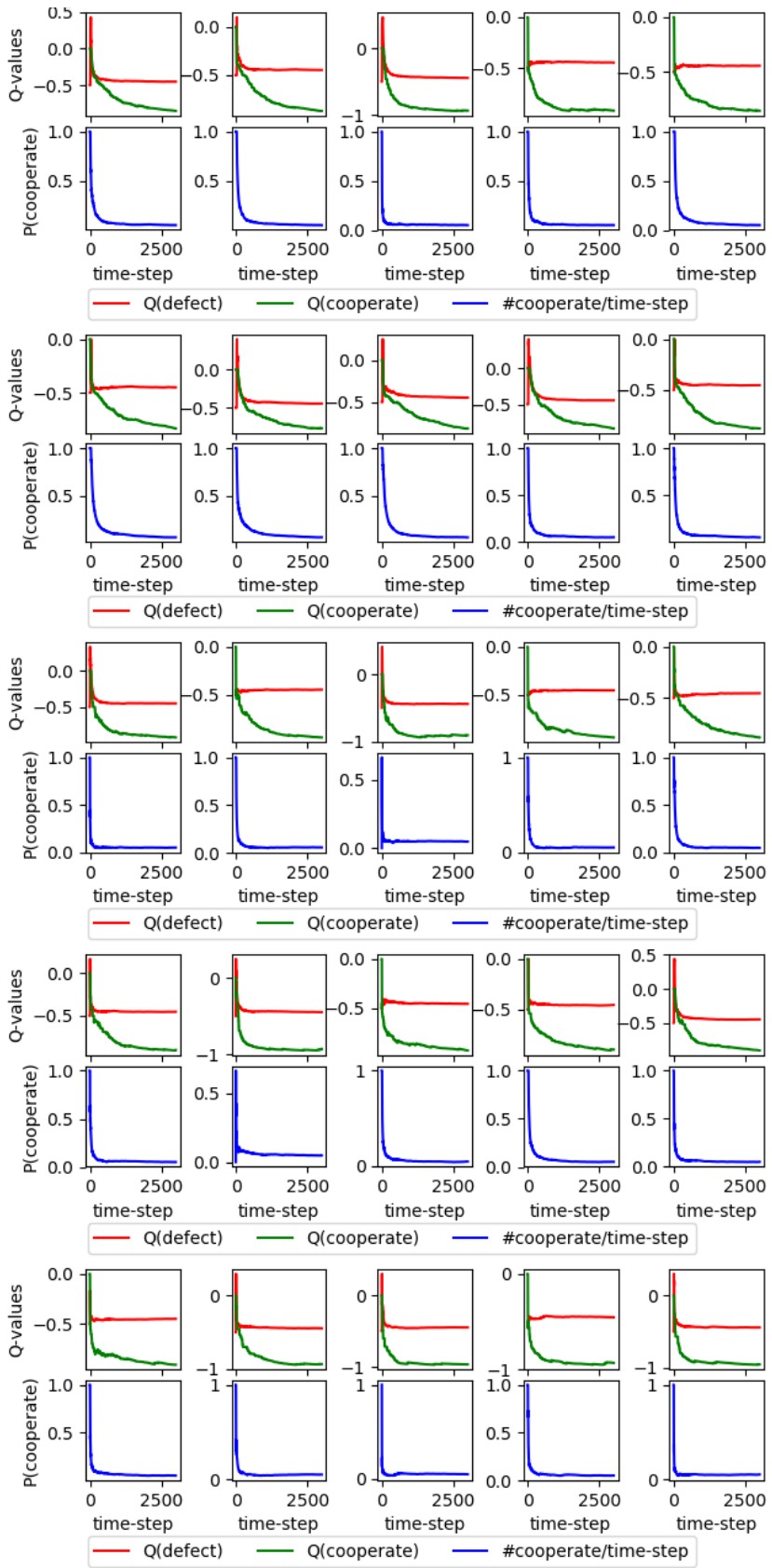

Figure 11: More independent experiments with Q-learning, exactly following Figure 5, except also using context swapping. This leads to a 100% success rate on the unit test.

## C.2 Content recommendation

### C.2.1 Environment details

The evironment has the following components:

1. **User type**, $x^t$: categorical variable representing different types of users. The content recommender conditions its predictions on the type of the current user.
2. **User loyalty**, $\mathbf{g}^t$: the propensity for users of each type to use the platform. User $x^t$ is sampled from a categorical distribution with parameters given by $\mathrm{softmax}(\mathbf{g}^t)$.
3. **Article type**, $y^t$: a categorical variable (one-hot encoding) representing the type of article selected by the user.
4. **User interests**, $\mathbf{W}^t$: a matrix whose entries $W^t_{x,y}$ represent the average interest user of type $x$ have in articles of type $y$.

At each time step $t$, a user $x^t$ is sampled from a categorical distribution (based on the loyalty of the different user types), then the recommendation system selects which type of article to present in the top position, and finally, the user selects an article. The goal of the recommendation system is to predict the likelihood that the user would click on each of the available articles, in order to select the one which is most interesting to the user.

User loyalty for $x^t$ then changes in accordance with the self-selection effect, increasing or decreasing proportionally to their interest in the top article. The interests of user type $x^t$ (represented by a column of $\mathbf{W}^t$) also change; in accordance with the illusory truth effect, their interest in the topic of the top article (as chosen by the recommender system) always increases. Overall, this environment is an extremely crude representation of reality, but it allows us to incorporate both the effects of self-selection (via covariate shift), and the illusory truth effect (via concept shift).

Formally, this environment is similar to a POMDP\R, i.e. a POMDP with no reward function, also known as a **world model** (Armstrong & O'Rourke, 2017; Hadfield-Menell et al., 2017); the difference is that the learner observes the input before acting and only observes the target after acting. The states, observations, and actions given below.

$$s^t = (\mathbf{g}^t, \mathbf{W}^t, x^t, y^t)$$

$$o^t_{\mathrm{pre}},\ a^t,\ o^t_{\mathrm{post}} = (x^t, \hat{y}^t, y^t)$$

The state transition function is defined by:

$$\mathbf{g}^{t+1}_{x^t} = \mathbf{g}^t_{x^t} + \alpha_1 W^t_{x^t, \hat{y}^t}$$

$$\mathbf{W}^{t+1/2}_{x^t, \hat{y}^t} = W^t_{x^t, \hat{y}^t} + \alpha_2; \quad \mathbf{W}^{t+1}_{x^t} = \frac{\mathbf{W}^{t+1/2}_{x^t}}{\|\mathbf{W}^{t+1/2}_{x^t}\|_2}$$

$$x^{t+1} \sim \mathrm{softmax}(\mathbf{g}^{t+1})$$

$$y^{t+1} \sim \mathrm{softmax}(\mathbf{W}^{t+1}_{x^{t+1}})$$

Where $\hat{y}^t$ is the top article as chosen by the recommender, and $\alpha_1$, $\alpha_2$ represent the rate of covariate and concept shift (respectively). The update for $\mathbf{W}^{t+1}$ merely increases the interest of user type $x^t$ in article type $\hat{y}^t$, then normalizes the interests for that user type.

### C.2.2 Reproducibility details

For these experiments, the recommendation system is a ReLU-MLP with 1 hidden layer of 100 units, trained via supervised learning with SGD (learning rate = 0.01) to predict which article a user will select. Actions are sampled from the MLP's predictive distribution. We apply PBT without any hyperparameter selection (this amounts to just doing the EXPLOIT step), and an interval of 10, selecting on accuracy. We use a population of 20 learners (whether applying PBT or not), and match random seeds for the trials with and without PBT. We initialize $\mathbf{g}^1$ and $\mathbf{W}^1$ to be the same across the 20 copies of the environment (i.e. the learners start with the same user population), but these values diverge throughout learning. For the environment, we set the number of user and article types both to 10. Initial user loyalties are randomly sampled from $\mathcal{N}(0, 0.03)$, $\alpha_1 = 0.03$, and $\alpha_2 = 0.003$.

### C.2.3 CONTEXT SWAPPING IN CONTENT RECOMMENDATION

We believe context swapping is not appropriate for the content recommendation environment, since when the environments diverge, optimal behavior may differ across environments. Nevertheless, we ran experiments with it for completeness. The main effect appears to be to hamper learning when PBT is not used, see Figure 12. Notably, it does not appear to significantly influence the rate or extent of ADS when combined with PBT.

### C.2.4 EXPLORATION OF ENVIRONMENT PARAMETERS

In Figure 13, we examine the effect of the rate-of-change parameters ($\alpha_1$, $\alpha_2$) of the content recommendation environment on the results provided in the paper. As noted there, our results are qualitatively consistent so long as (1) the initial user distribution is approximately uniform, and (2) the covariate shift rate ($\alpha_1$) is faster than the concept shift rate ($\alpha_2$). These distributions are updated by different mechanisms, and are not directly comparable. Concept shift changes the task more radically, requiring a learner to change its predictions, rather than just become accurate on a wider range of inputs. We conjecture that changes in $P(y|x)$ must therefore be kept smooth enough for the outer loop to have pressure to capitalize on ADS.

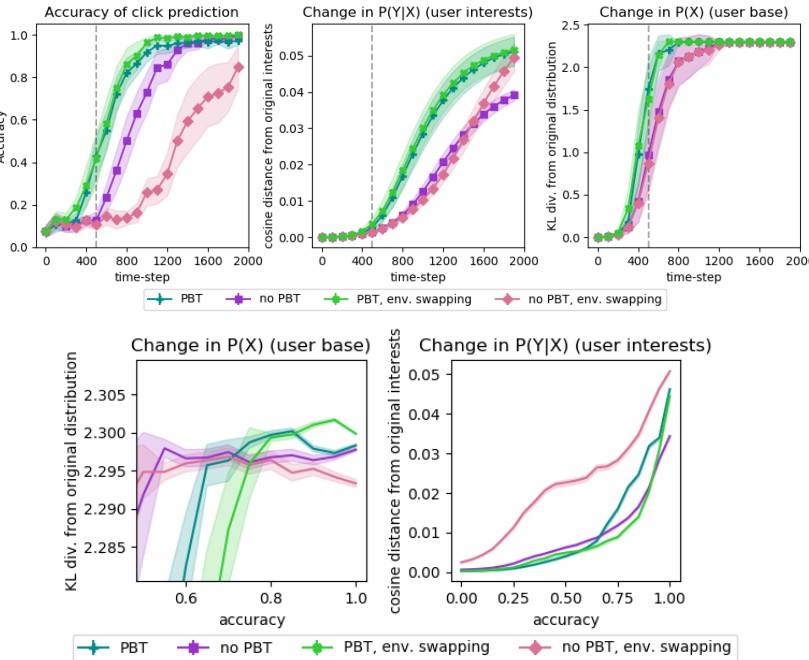

Figure 12: Context swapping doesn't have the desired effect in the content recommendation environment.

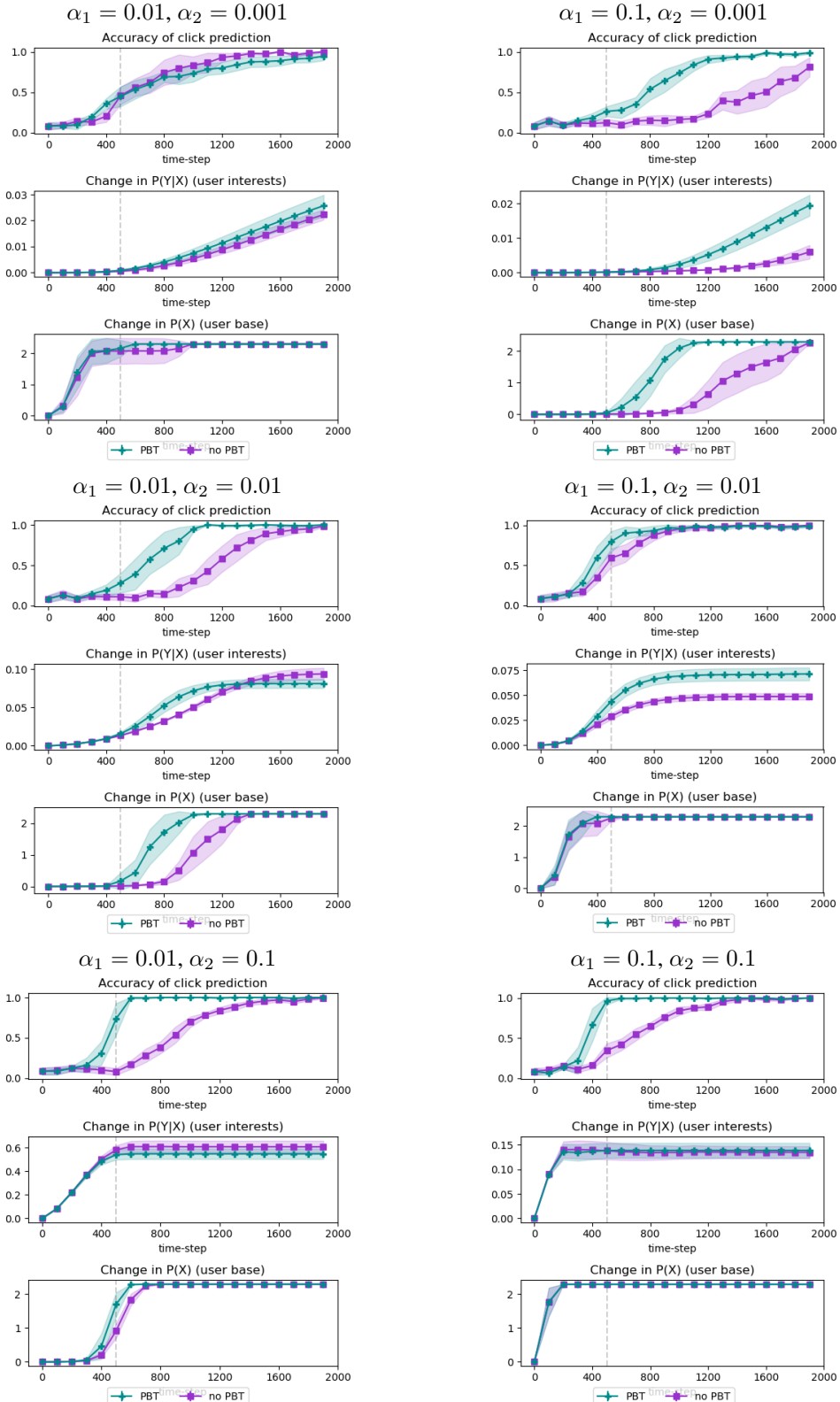

Figure 13: Content recommendation results for different values of $\alpha_1$, $\alpha_2$.

