# OpenReview forum: "Revealing the Incentive to Cause Distributional Shift"
_ICLR.cc/2022/Conference — ICLR 2022 Submitted_

### Official Review · Reviewer_vVp6 · 2021-11-01

**Correctness:** 4
**Technical Novelty And Significance:** 3
**Empirical Novelty And Significance:** 2
**Recommendation:** 3
**Confidence:** 4

**Main Review:**

Overall, I like the problem studied in the paper but I have two concerns about related work and the role of these unit tests for informing you about how to design algorithms.

**Related work**

I take some issue with the authors reintroducing an existing conceptual idea under a different name. The realization that models can induce distribution shifts is not novel and it has for example previously been conceptualized in the context of risk minimization at a similar level of abstraction under the term *performative shift*.

Let me elaborate on this: I am not sure the authors are aware of the very related literature on performative prediction. Perdomo et al [ICML’20] define performative effects as a shift in distribution induced by the predictive model you deploy. Then, performative prediction extends classical supervised learning and formalized as a risk minimization problem over a model dependent distribution. The goal is to minimize the loss over the distribution that will surface from deploying the model. In this context you can think of the loss as measuring the accuracy of your model, but you could also think of it as a general utility function on the induced distribution.

I don’t understand how the conceptual idea of performative distribution shifts is different of what you newly introduce as “auto-induced distribution shift”. Introducing a new term for something that already exists is not a good practice. At least you should discuss how it is different and why it needs a new term, or otherwise simply adopt an existing term.

Follow-up works of Perdomo et al have studied both, algorithms that are performatively-aware in that they incorporate the shift into their updates, and naïve retraining strategies that are oblivious to the shifts and myopically optimize on the current distribution. In line with your findings, it has been realized that this leads to fundamentally different solution concepts. I really think connecting more to this line of work would be very interesting and beneficial for both communities. I will mention a couple of references in the end.

**The role of the unit tests**

 think the focus of this work on the incentives and strategies that the learner pursues to optimize its utility is very interesting. However I am not fully understanding what you hope the general conclusions from the outcome of your unit tests would be.

The simple unit-tests are designed to analyze an algorithm’s behavior in the presence of undesirable incentives to shift the distribution. These examples are simple enough so that algorithm behavior is easy to interpret on this example and allows for a comparison between algorithms. It is interesting, although not that surprising that the behavior differs across algorithms.

The term unit tests implies that you want good algorithms to pass it. So this would mean you blame them for finding this alternate solution. Is that what your intention is? Do you think you can avoid this behavior without hurting the performance of the algorithm along other dimensions? As you mentioned in the introduction it does not seem to be a problem of the algorithm, but a problem specification issue.

In your examples the prediction task can be solved myopically and specification gaming requires taking observations of induced distributions into account. So what you will figure out with your test is whether your algorithm can learn from this feedback or not. Are you advocating for algorithms be 'bad' if they can do this? Would be nice if you could elaborate more on how you think these tests could help for certifying or improving your algorithms.

I have a few additional questions I am curious about:
-	Do you think that in general specification gaming leads to larger (or worse) shifts than myopic algorithms? even if you optimize myopically you will inevitably cause shifts that will drive the distribution to a different state over time.
-	It seems to me that meta-learning has more options to detect such actions, but it can only take advantage of this if the different configurations are exploring sufficiently. Is this correct?
-	You mention that it is hard to specify an objective to anticipate such shifts. If you are aware of these shifts, would it be possible to use constraints or some sort of regularization to bias the solution towards a desirable induced distribution?
-	It is not clear what Figure 4 (top row) is showing- what is $\|y_2\|$ referring to?

Some references:

- *Perdomo et al.* Performative prediction. 2020
- *Miller et al.* Outside the Echo Chamber: Optimizing the Performative Risk. 2021
- *Izzo et al.* How to learn when data reacts to your model: performative gradient descent. 2021
- *Brown et al.* Performative Prediction in a Stateful World. 2020

**Summary Of The Paper:**

The paper realizes that the ability of a model to induce a distribution shift can provide incentives to the learner that are not necessarily in line with the intention of the algorithm designer. It might find other means to reduce the risk through specification gaming rather than improving predictions. The authors propose two simple unit tests to detect such behavior and illustrate in these examples that meta-learning makes algorithms more susceptible to this potentially unanticipated behavior. As an approach to partially fix the issue they propose context-swapping such that the algorithm does not directly see the consequences of its actions.

**Summary Of The Review:**

I think the incentives perspective offered by this paper on model-induced distribution shifts is interesting. Designing simple tests enables the authors to provide some interesting insights about the relative performance of algorithms. However I am not sure what the algorithm designer can learn from the results of these unit tests. My second concern is the treatment of related work. I don’t think you should claim the definition of this phenomena as a novel contribution.

I am willing to increase my score if the authors provide a satisfactory solution of how to address the conceptual overlap with existing work and they can convince me about the usefulness of these unit tests to guide algorithm design.

---

### Official Review · Reviewer_yBdS · 2021-11-01

**Correctness:** 4
**Technical Novelty And Significance:** 2
**Empirical Novelty And Significance:** 2
**Recommendation:** 3
**Confidence:** 3

**Main Review:**

Strengths:
- the topic of the paper is important and interesting. In today's world where ML algorithms are deployed on humans, it is a stretch to assume i.i.d. or static user data.
- there is novelty in the paper in that the goal is not necessarily to deal with or compensate for ADS, like in much related work, but rather to try to detect incentives for the learner to induce ADS.

Weaknesses:
- this seems very, very related to the idea of strategic behavior in machine learning, and of "Performative Prediction" (Juan C. Perdomo, Tijana Zrnic, Celestine Mendler-Dünner, Moritz Hardt). I could not find this line of work in the citations of the paper, and I think it may need to be there. Currently, the paper ignores an important line of work that considers exactly the same effects (though maybe in a slightly different setting(.
- I do not think the main insights of the paper are super surprising or new. I don't think there is really anything novel in observing that PBT reveals incentives better -- obviously PBT will do "well" simply through the fact that it heavily incentivizes exploration. It is also not surprising that some ML algorithms can learn to reveal this incentive even if they look at myopic utilities: for example, imagine a 2-player game in which one player (so here it could the learner in the previous time step) does best response and the other one (the learner in the current time step) plays a no-regret algorithm (but the loss function and the update rule at a given round only depends on the utility in this round, and not the future utilities -- for example multiplicative weights); then it is known that such an algorithm will converge to an equilibrium of the game, which means in the prisoner's dilemma it would discover the incentive to cooperate.
- The paper reads like a (still relatively interesting) collection of examples rather than a "unified" research paper. I think the paper would gain from having a unified model for ADS that all the current examples in the paper derive from. It would also be useful to have a centralized definition for the unit tests it uses for example. Right now, I find the unit tests very ad-hoc and case-by-case, and I think the paper would benefit from defining them in a more principled way.

**Summary Of The Paper:**

The paper looks at the problem of auto-induced distributional shift, which is what happens when the inputs to a machine learning algorithm are affected by the algorithm used for learning itself. In general, ML algorithms do not face an i.i.d. distribution of data or users; rather, in recommendation systems for example, the choice of content displayed will affect how users react to it and even which users we face. In machine learning applied to users or humans, said users/humans we make decisions on may try to react to the decision rule in order to obtain better outcomes. The authors aim to study ADS, and to design tests for incentives, to detect whether given algorithms hide or detect incentives that cause ADS.

**Summary Of The Review:**

I think the paper has some interesting ideas, but I think it needs to be improved before it is above the bar for acceptance at ICLR.

---

### Official Review · Reviewer_nHgz · 2021-11-03

**Correctness:** 4
**Technical Novelty And Significance:** 2
**Empirical Novelty And Significance:** 2
**Recommendation:** 3
**Confidence:** 2

**Main Review:**

 - I could be wrong, but this paper to me is but an empirical extension to Performative Prediction (2020). Performative Prediction (2020) studies Repeated Risk Minimization, which to me is PBT (based on the author's description) but with only one learner; there is no parameter perturbing and replacing going on, only the retraining is being performed. As expected, there is a distribution shift when you do Repeated Risk Minimization. With PBT, we now have multiple learners doing the same thing. I don't find that PBT induces distribution shift surprising at all. The utility function of the data studied in this paper is more complicated than that in Performative Prediction (2020), but I don't see it matters for the main findings of this paper.

 - We, the machine learning engineers, are meta-learners. We tune parameters of models, deploy them in the field, and observe distribution shifts. The main message of this paper, in my opinion, did not need to be motivated by meta-learning or PBT at all; the message could have been made more powerful without it. PBT can be added in the experiments as a way to simulated a meta-learner, i.e. human.

 - For a paper that uses game theory terminology in its title, I find it hard to follow its game theory elements. The necessary notations are introduced very late, in Sec. 5.2. There is also an overuse of italic fonts, which have me wondering if those are concepts that the authors will define concretely using notations later in the article (which they aren't).

 - "Revealed incentive" sounds odd. In game theory, the agents' incentives are well defined: the classifier wants higher accuracy, and the data wants, say, better movie recommendations. The game is then played, and the equilibrium is then observed. One can argue that the classifier causing a distribution shift is an unwanted equilibrium of this game, but saying that the change of the data's equilibrium strategy in this sequential game is an "incentive" of the classifier being "revealed" is ... not standard.

 - Typo: Captions to Figure 2: state s, action s, reward s tuples.

**Summary Of The Paper:**

This paper proposes to study Auto-induced Distribution Shift (ADS), the phenomenon that models can create a feedback-loop: the prediction of a model influence user behaviors when it is deployed and retrained iteratively, which, in turn, affects the accuracy measure of the model. The paper empirically shows that a meta-learning algorithm called PBT causes a distribution shift instead of maximizing accuracy.

**Summary Of The Review:**

I believe this paper is an empirical, meta-learning extension to Performative Prediction (2020). I believe the main findings of the paper are not attributed to meta-learning, but to the idea of repeated retraining for online learning, which is well explored in Performative Prediction (2020). In addition, this paper does not cite Performative Prediction (2020), and the writing (especially the use of notations and terminologies) needs work. Therefore, I incline to recommend rejection.

---

### Official Review · Reviewer_C74h · 2021-11-05

**Correctness:** 3
**Technical Novelty And Significance:** 2
**Empirical Novelty And Significance:** 2
**Recommendation:** 5
**Confidence:** 3

**Main Review:**

The paper is rich in content and references. I like the idea of using POMDP to simulate the characteristics of a self-selection loop, as well as the novel adaptation of the prisoner's dilemma. There are other interesting ideas like the close examination of meta-learning and q-learning.

However, my main concern of this paper is the lack of validation in real-world scenarios. For example, could the authors collect data to match the POMDP simulations with real-world events? As the paper proposes tools to study the techno-societal impacts of technology, having empirical validation seems like an important step. The two simulations in the paper alone do not show clear evidence of the generality of the proposal.

This concern is compounded with my lack of intuition of why we must "hide the incentives" or, as I understand it, completely block the gradient path, between a state change and the final reward. As the authors suggest as well, having cooperation between the environment states and the agent actions is important in many POMDP problems. For example, we cannot ask a recommender system to not have an influence on the society. Rather, we should be asking how a recommender system should influence the society. I.e., we may want more interpretability in the information leakage than a complete blockage. The oversimplification of the problem leads to lesser impact.

On the technical side, I have some confusions on Experiment 1. It seems that the agents never get into the "cooperate" state with more than 50% chance in global probability, when the gradient path is blocked between the current and the future states. It becomes hard to tell whether ADS happened or rather the agent model simply failed to converge. Is it possible to produce near-100% "cooperate" rates in the given setting? I also have problems following the explanations for the "cooperate" result. It seems that this is due to some kind of gradient leakage in the outer loop of meta-learning, but I am not sure. I also do not have strong intuitions in the analysis of the q-learning algorithm.

Experiment 2 is pretty much self-evident. In the pursuit of optimal strategies, the agent policy, as well as the state visitation, will eventually saturate around the optimal choices. This concentration effect is exactly why a mountain car agent can successfully move up the valley through a set of particular actions to receive the highest rewards, given sufficient reinforcements. In RecSys, I would not completely dismiss the concentration effects. For example, in product recommendation, the concentration leads to efficient goal-oriented shopping experience. On the other hand, for content recommendation, the concentration could be bad - but this is often compensated by the uncertainty of new contents ingested in real-time. As a result, people still consider Fox/CNN to be valuable sources of information despite their potential biases. I would love to see more discussions on the complex nature of the bias problem.

Some style comments:
- [abstract] "whether an algorithm will hide or reveal incentives" - Missing a lot of contexts to understand this sentence on first pass.
- [abstract] "correctly diagnoses its propensity to reveal incentives to ADS" - Seems like an over-claim.
- [intro] "2. Changing the distribution of users such that predictions are easier to make" - RecSys do not change the distribution to make the predictions easier to make. They just happen coincidentally.
- [intro paragraph 2 middle] "quantify the desirability of all possible means" - Means and meant in a few lines above cause confusions.
- [page 2 paragraph 3] "one that does not induce ADS; the other is hidden and we want it to remain hidden" - fragmented sentence.
- [contribution Point 2] "liable" - replace with simple words.
- [contribution Point 3] "confirm qualitative features" - hard to parse.
- [Page 3 PBT] "because they give the OL more control than many other meta-learning algorithms" - clarify what other meta-learning algorithms this is referring to.
- [Page 3 second-to-last paragraph] "self-refuting predictions" - use simple words
- [Page 3 second-to-last paragraph] "Leike et al., 2018" - does not seem to talk about news manipulation
- [Page 3 last paragraph] - there is a lot of missing context. what is i.i.d.? how is RL environment different from the i.i.d. assumption?
- [Section 4 first paragraph] "the incentive to drive users away" - there is never an incentive to drive users away in a RecSys. It just happens.
- [Page 4 second to last paragraph] "of influencing" - to influence
- [Page 4 last paragraph] "Everitt & Hutter 2018" - 2019.
- [Page 7 (2) exploiting correlation] - I do not quite follow this paragraph
- [Page 8 Section 5.2.1] "the covariate shift rate (alpha_1) is faster than the concept shift rate (alpha_2)" - are they defined earlier?
- [Section 5.2.1] - where are the results for context switching?
- [Page 9 Safety and incentives] "the efficacy of this approach of hiding incentives and ways in which it can fail" - too many concepts in one sentence. not sure what the "ways to fail" refer to.
- [Page 9 Conclusion second paragraph] "affect their propensity to reveal incentives for ADS" - propensity => likelihood

Edit:
I second Reviewer yBdS in the suggestion that no-regret algorithms in the outer loop could in principle "memorize" the aggregated information from the inner loop.

**Summary Of The Paper:**

The paper talks about a self-selection phenomenon which the authors call Auto-induced Distributional Shift (ADS). This phenomenon is common in recommender systems, where the promotion of a type of contents (e.g., liberal versus conservative) leads to a change in the active user base. Self-selection bias is a big topic and hard to quantify precisely. Instead, the paper makes a novel connection to Partially-Observable Markov Decision Process (POMDP), where the self-selection effect is analogous to the changes in the environment states (i.e., the active user base) after agent decisions (i.e., content recommendation). With this setup, the paper is able to simulate the self-selection process and analyze the system parameters that may affect the characteristics of the process, such as its mixing time.

The paper also contains some other discussions on whether we can "hide the incentives" or, as I understand it, completely block the gradient path, between a state change and the final reward as a way to prevent ADS. This is substantiated by a self-cooperating/defecting game adapted from prisoner's dilemma and a comparative study of a few learning algorithms including meta-learning and q-learning. This seems to bring novelty, but I have some questions about the technical details as well as how to apply the ideas in more general cases.

**Summary Of The Review:**

I like the idea to simulate self-selection bias with POMDP. However, I have questions in the discussions of the comparative studies (ADS effects as the authors explained versus simple failures of convergence). I also question the proposal to completely block the gradients to discourage taking advantage of any changes in the environmental states. I gave counter examples in RL mountain car problem, goal-oriented online shopping, and even news recommendation when opinion biases do not stop it from providing useful information. The paper's proposal seems oversimplified and it needs to be validated in real-world scenarios to produce sufficient impacts for this venue.

---

### Decision · Program_Chairs · 2022-01-20

**Decision:**

Reject

**Comment:**

This paper proposes to study Auto-induced Distribution Shift (ADS), the phenomenon that models can create a feedback-loop: the predictions of a model influence user behaviors when it is deployed, which, in turn, affects the accuracy measure of the model. The paper empirically shows that a meta-learning algorithm called PBT causes a distribution shift instead of maximizing accuracy. While the premise of this paper is interesting, the proposed frameworks are very similar to the idea of strategic behavior in machine learning, and of "Performative Prediction" (Juan C. Perdomo, Tijana Zrnic, Celestine Mendler-Dünner, Moritz Hardt). However, this line of work is neither cited nor discussed in this paper. In addition, the paper is hard to read in certain parts. We encourage the authors to compare their work with performative prediction. We hope the authors find the reviews helpful.